# In the Eye of the Storm: A Quantitative and Qualitative Account of the Impact of the COVID-19 Pandemic on Dutch Home Healthcare

**DOI:** 10.3390/ijerph19042252

**Published:** 2022-02-16

**Authors:** Anne O. E. van den Bulck, Maud H. de Korte, Silke F. Metzelthin, Arianne M. J. Elissen, Irma H. J. Everink, Dirk Ruwaard, Misja C. Mikkers

**Affiliations:** 1Department of Health Services Research, Faculty of Health, Medicine and Life Sciences, Care and Public Health Research Institute (CAPHRI), Maastricht University, 6200 MD Maastricht, The Netherlands; s.metzelthin@maastrichtuniversity.nl (S.F.M.); a.elissen@maastrichtuniversity.nl (A.M.J.E.); i.everink@maastrichtuniversity.nl (I.H.J.E.); d.ruwaard@maastrichtuniversity.nl (D.R.); 2Department of Economics, Tilburg University, 5037 AB Tilburg, The Netherlands; mkorte@nza.nl (M.H.d.K.); mmikkers@nza.nl (M.C.M.); 3Dutch Healthcare Authority (NZa), 3502 GA Utrecht, The Netherlands; 4Tilburg Law and Economics Center (TILEC), Tilburg University, 5000 LE Tilburg, The Netherlands

**Keywords:** home care services, COVID-19 pandemic, management, nurses, patients, descriptive statistics, focus groups

## Abstract

The COVID-19 pandemic has severely affected healthcare delivery across the world. However, little is known about COVID-19’s impact on home healthcare (HHC) services. Our study aimed to: (1) describe the changes in volume and intensity of HHC services and the crisis management policies implemented; (2) understand the responses and the experiences of HHC staff and clients. We conducted an explanatory sequential mixed methods study. First, retrospective client data (N = 43,495) from four Dutch HHC organizations was analyzed. Second, four focus group interviews were conducted for the strategic, tactical, operational, and client levels of the four HHC organizations. Our results showed that both the supply of and demand for Dutch HHC decreased considerably, especially during the first wave (March–June 2020). This was due to factors such as fear of infection, anticipation of a high demand for COVID-19-related care from the hospital sector, and lack of personal protective equipment. The top-down management style initially applied made way for a more bottom-up approach in the second wave (July 2020–January 2021). Experiences vary between levels and waves. HHC organizations need more responsive protocols to prevent such radical scaling-back of HHC in future crises, and interventions to help HHC professionals cope with crisis situations.

## 1. Introduction

The COVID-19 pandemic has led to radical changes in healthcare delivery across the world since its onset in the beginning of 2020 [1]. Uncertainty about the course and consequences of the disease, the risks of transmission, uncertainty about the effectiveness of personal protective equipment (PPE), and the lack of PPE for healthcare professionals all resulted in drops in the volumes of regular, non-COVID-19-related care provided [2]. This has affected not only hospital settings, but also the home healthcare (HHC) sector.

In the Netherlands, HHC can be defined as ‘formal nursing services and personal care provided by HHC professionals in clients’ own homes’ [3]. Nursing services can be of a technical, supportive, rehabilitative, or preventive nature. Personal care services relate to assistance with activities that are part of daily living, such as dressing, eating, and washing [3]. Different types of care can be delivered to various types of patients, such as the chronically ill, disabled people, older adults, and people at the end of their lives. HHC encompasses both long-term care at home and short-term care at home—following discharge from hospital, for instance. HHC in the Netherlands is provided mainly by: (1) registered nurses with either a Bachelor’s degree or a senior secondary vocational education (i.e., European Qualifications Framework (EQF) educational level 6 or educational level 4, respectively); (2) certified nursing assistants (EQF educational level 3); and (3) care assistants. District nurses (registered nurses with a Bachelor’s degree) are responsible for formally assessing care needs for services covered by the Dutch Health Insurance Act (HIA), taking account of clients’ self-reliance and the resources available in their social network. The majority of Dutch HHC providers are not-for-profit organizations, acting in a competitive environment in which the number of commercial providers is growing [4]. In 2018, approximately 3070 HHC organizations, including self-employed nurses, delivered services to more than 580,000 clients in the Netherlands [5]. Since 2015, the HIA has made health insurance compulsory and covers essential healthcare services, including HHC for clients who need care less than 24 h per day [6].

Compared to hospitals, the HHC sector received much less guidance and protocols with respect to how to provide care safely during the COVID-19 pandemic. To make matters worse, only limited quantities of facemasks, disinfectants, or other PPE supplies were distributed to HHC professionals [7]. Research has shown that during the first months of COVID-19 infections, HHC professionals felt alone, undervalued, and isolated [8]. This was a period when HHC organizations were focusing mainly on minimizing the risk of transmitting the virus to frail clients and their caregivers (whether formal or informal). Another priority was to relieve hospitals by ensuring that post-COVID-19 patients could be cared for at home [7]. This resulted in the cancellation of much HHC that was regarded non-essential or non-urgent [9,10]. However, little research has been conducted on the extent to which the volume and intensity of HHC services changed over the course of the pandemic. Neither do we know which factors determined these changes, i.e., how did HHC organizations and professionals decide which types of clients or HHC services to prioritize, and which alternatives—such as e-health or informal care—to offer? Even though these factors could be attributed to client characteristics or the availability of informal care, it is likely that regional or national policy also played an important role [11].

On the one hand, the scaling-back of HHC services had a major impact on clients and informal caregivers, such as family members providing care to loved ones. For instance, research has shown that the care burden of informal caregivers increased during the first months of the pandemic, as they often had to replace care that had previously been provided by HHC professionals [10]. On the other hand, even though changes in the volume and intensity of HHC provision during the pandemic were generally unwanted and viewed negatively, they could also provide some unique learning opportunities for the future of HHC. For instance, alternative means of providing HHC—such as e-health or tools that encourage clients to perform certain care tasks independently—could help to keep HHC sustainable in the future. We conducted this study to understand and learn lessons from the behaviors and experiences of those who have been providing, managing, and receiving HHC during a period of crisis. Our first aim was to describe how Dutch HHC organizations responded to the COVID-19 pandemic (March 2020–January 2021), in terms of changes in the volume and intensity of HHC services, and the crisis management policies implemented. Second, we aimed to understand these responses at all levels of HHC—i.e., strategic, tactical, operational, and client levels—and to learn more about their experiences. These aims will be addressed using an explanatory sequential design.

## 2. Materials and Methods

### 2.1. Design

This study was conducted in the Netherlands between March 2020 and January 2021, thus covering the first two waves of COVID-19 infections in the Netherlands. The first wave was between March and June 2020, and the second wave between July 2020 and January 2021. The study is based on a mixed methods design, and more specifically an explanatory sequential design consisting of two phases [12]. The quantitative findings based on descriptive analyses in the first phase were explained by the qualitative findings from focus group interviews in the second phase. The objective of quantitative methodology in the first phase was to describe the changes in volume and intensity of HHC services due to the COVID-19 pandemic, herewith corresponding mainly to the first study aim. For the qualitative methodology in the second phase, the objectives were to understand these changes identified in the first phase, to describe crisis management policies implemented, and to learn about the experiences of providing and receiving HHC during the crisis. This corresponds to both the first and second study aims.

### 2.2. First Phase: Quantitative Methodology

#### 2.2.1. Sampling and Data Collection

Four Dutch HHC organizations participated in this study. They were selected based on their participation in a pilot study conducted previously regarding the development of case-mix classification for HHC in the Netherlands [13]. The selected HHC organizations operate in various regions of the Netherlands and provide services to relatively large client populations (i.e., about 2000–4000 clients per organization at any given time).

For the first phase of this study, routinely collected, electronic health records (EHRs) were used to extract retrospective client data for the four participating Dutch HHC organizations. All clients who received HHC (financed through the HIA) from the HHC organizations were involved in the study. No exclusion criteria were applied in the selection of clients whose data were used for our study. The data used contained the per day registration of formal HHC hours provided to each client (indicated by an anonymous client ID) between June 2019 (i.e., the start date of the previously conducted pilot study) and January 2021. In total, the data included 43,495 unique clients for this period, with an average of 11,528 unique clients per month.

#### 2.2.2. Data Analysis

The quantitative data were analyzed using R. Descriptive statistics were used to determine and visualize trends in the care delivery and flow of clients during the COVID-19 pandemic in the period March 2020–January 2021, and compared to the previous year. Since the data from the previous period (without exposure to COVID-19) were only available from June 2019, comparison to the COVID-19 pandemic was only possible from that month onwards (i.e., from week 23 of 2020 onwards; see Appendix A ‘Analyses on trends in HHC use during the COVID-19 pandemic on Dutch HHC claims data’ for an additional analysis on more aggregated claims data for which complete years 2019 and 2020 were available). Care delivery was operationalized using the parameters of total HHC hours utilized and total number of clients receiving HHC, summed for the four HHC organizations. To compare the trends before and after the start of the pandemic, we calculated the average number of hours of HHC utilization and the number of HHC clients in weeks 2–10 of 2020. In this way, we could express the numbers in the weeks after that as percentage differences from this average. For clients who stopped receiving care, we calculated the number of clients not utilizing HHC for 40 days or longer and defined these clients as no longer receiving HHC. We also analyzed the trend in the intensity of HHC use per client. Intensity was defined as the total hours of HHC used by a client every week. We assigned the following categories: 0–0.5 h, 0.5–2 h, 2–4 h, 4–8 h, and more than 8 h of HHC use per week.

### 2.3. Second Phase: Qualitative Methodology

#### 2.3.1. Participants

In the second phase, a sample of stakeholders from various levels was recruited among the four HHC organizations: (a) strategic level (i.e., member of board of directors of strategic manager); (b) tactical level (i.e., operational managers or policy makers); (c) operational level (i.e., district nurses); (d) client level (i.e., client council members). Four focus groups, each composed of members from all four HHC organizations, were established based on stakeholder level to ensure that there was room for open communication. The aim was to recruit at least four participants per focus group with a maximum of 12 participants [14], which implied a maximum of three participants per organization. Potential participants for each level were selected by the HHC organization. Subsequently, the participants were invited to take part in the study by e-mail. If they were interested in participating, they received a second email including an information letter and informed consent form.

We contacted between 6 and 12 participants for each level. The total numbers of participants in each focus group interview ranged between 4 and 11 (see Table 1 for the number of participants per level and their functions). For each level, all four organizations were represented by at least one participant.

#### 2.3.2. Data Collection

Four semi-structured focus group interviews were conducted in November and December 2020. The interviews were led by two researchers (AvdB, MdK). After a short introduction, the quantitative results of the first phase were presented to the participating stakeholders followed by a semi-structured interview. For the interviews, a topic list was used covering the following topics: (a) changes in the use of HHC (i.e., number of clients, type of HHC, and amount of HHC); (b) explanations for these changes from the four perspectives (see Table 1) as well as the regional/national context; and (c) crisis management policy (i.e., what was done, how was it determined, and how was it viewed). Due to the pandemic, the focus group interviews were held using the online video conferencing tool Zoom. The interviews took a maximum of two hours and were audio-recorded using a voice recorder.

#### 2.3.3. Data Analysis

The focus group interviews were transcribed and anonymized by replacing all the names in transcripts with a code. An analysis of the interview transcripts was carried out using Atlas.ti to help the researcher understand, organize, and interpret the data. The first step in the coding was to apply a combination of deductive coding, using a coding scheme based on research aims, and inductive coding. This was completed by three researchers (AvdB, MdK, IE) for the first interview. Next, one researcher (MdK) coded the remaining interviews. In the second step, a thematic analysis [15] of the coded quotations was conducted by pairs of researchers: MdK and MM focused on changes in volume and intensity and possible explanations, AvdB and AE analyzed the data on crisis management policies and experiences, and finally MdK and AvdB studied the quotations resulting from the inductive coding. Throughout the analysis, we were interested in determining similarities and differences between findings on the different levels, i.e., the strategic, tactical, operational, and client levels. Therefore, data saturation on the identification of themes of findings was determined across the focus group interviews. The main findings are presented for each theme, supported by quotes.

### 2.4. Data Management

The quantitative data was transferred from the participating HHC organizations to the Dutch Healthcare Authority using a secure route and was only accessible to the researchers involved at the Dutch Healthcare Authority. The quantitative data was anonymized by using a unique client identifier rather than a name or social security number. The data was stored on the secure network drive of the Dutch Healthcare Authority and will be archived by the Dutch Healthcare Authority for a period of at least 10 years after the publication of this study.

The qualitative data was stored in a folder on the secured Maastricht University server with no intermediate cloud storage. All recordings were labeled with a unique ID that corresponds to the date on which the focus group took place and the name of the relevant level, and therefore includes no information that could be used to trace individual participants. The data was only accessible by the researchers involved at Maastricht University and the Dutch Healthcare Authority. The data will be archived at Maastricht University for a period of at least 10 years after publication.

### 2.5. Ethics

Ethical approval for this study was granted by the medical ethical committee FHML-REC of Maastricht University. The research was classified as not subject to the Dutch Medical Research Involving Human Subjects Act (i.e., WMO) (reference number: FHML-REC/2020/112).

The quantitative data was processed without the explicit consent of participants on the basis of the Dutch Healthcare Authority’s legal obligation to supervise healthcare markets (article 16(a), Healthcare Market Regulation Act). Any information that could be used to identify individual persons was removed or anonymized prior to data collection. Participating organizations were provided with materials to inform clients about the purpose of this study in accordance with the General Data Protection Regulation. An opt-out form was created to allow nurses to flag clients who did not want to participate. 

Participants in the focus groups were sent an information letter and an informed consent form by email. Participation in the study was completely voluntary and withdrawal was possible at any moment prior to, during, or after the focus group, with or without providing a reason, by contacting one of the researchers. In the informed consent statement described, the participants agreed that the focus group would be recorded as they started the discussion. They were again asked for their consent at the start of each focus group. At all times, the participants were able to ask any questions they might have. No reimbursement, remuneration, or reward was given for participating in the study.

## 3. Results

The first part of the results section will focus on describing to what extent the volumes and intensity of HHC changed due to the COVID-19 pandemic, which was identified from the quantitative data. The second part of the results section will focus on understanding the responses from HHC staff and clients regarding the COVID-19 pandemic, and learning about their experience of providing or receiving HHC during the crisis.

### 3.1. Impact of the Pandemic on HHC Utilization

Figure 1, Figure 2, Figure 3 and Figure 4 show the changes in HHC utilization and the overall numbers of clients. In all the figures, two vertical dotted lines show the key moments at the beginning of the COVID-19 pandemic in the Netherlands. The black dotted line indicates the moment when the Dutch government first announced the start of preventive measures of COVID-19 (i.e., 12 March 2020). The grey dotted line indicates the closure of nursing homes to visitors and other non-essential persons, announced on 19 March 2020. Overall, the participants in the focus group interviews confirmed the quantitative findings that are described in the following paragraphs (note: in cases where their view differed, this is indicated clearly).

#### 3.1.1. Changes in Hours of HHC Utilized

The blue line in Figure 1 shows the trend in the total hours of HHC provided by the four HHC organizations in the year 2020 and the beginning of 2021. We see a decrease in HHC utilization of 13.4% in week 13 and 19% in week 17 compared to the preceding weeks. After week 18, the total hours of HHC increased slowly again. On average, the HHC organizations provided 5.7% fewer hours of HHC from week 23 until the end of the 2020 (shown by the blue line) compared to the same weeks in 2019 (shown by the brown line). The sudden decline and increase (in both years) around week 52 is due to the Christmas holidays.

#### 3.1.2. Changes in the Number of HHC Clients

The total number of clients using HHC decreased by 10.3% in week 13 and by 13.4% in week 17 compared to the average for the preceding weeks, as shown by the blue line in Figure 2. After week 18, the number of clients increased again, almost returning to the level of 2019 (i.e., the brown line) in the subsequent weeks.

Both the trend in the total HHC provided (Figure 1) and number of clients (Figure 2) (see Appendix A: Average HHC hours per client) appear to show that the average number of HHC hours per client decreased following the start of the pandemic and remained lower until at least January 2021.

Figure 3 shows the trend in the number of HHC clients who stopped receiving homecare, which peaked between weeks 11 and 14. After that, the usual (according to the participants) and somewhat unpredictable trend resumed. According to the participants, it is likely that the increase in the number of clients no longer receiving care after the start of the pandemic was an underestimate. For example, clients who themselves decided to stop receiving HHC due to the pandemic would still be checked on by district nurses by telephone or video call, but these checks were still registered as HHC hours so these clients were not counted as having stopped receiving homecare.

#### 3.1.3. Changes in HHC Intensity of Clients

In Figure 4, we see an increase in the number of clients receiving 0–0.5 h HHC per week after the start of the pandemic, and a decrease in HHC hours in the categories requiring heavier care. These trends did not seem to occur in the second wave of the pandemic. The heaviest category of care (over 8 h HHC per week) remained stable overall. 

Clients utilizing less HHC after the start of the pandemic often moved down by just one category, such as from ‘4–8 h’ to ‘2–4 h’ (see Appendix A: Changes in HHC intensity per client). In addition, there were some clients who actually received more HHC after the start of the pandemic.

### 3.2. Providing and Receiving HHC during a Crisis

The qualitative analysis focused on understanding the changes in HHC utilization throughout the first and second waves of the COVID-19 pandemic. Based on our analysis of the causes, consequences, and experiences, we were able to identify three main themes: (1) at home, alone? (2) managing uncertain demands; (3) being a nurse in times of crisis. Each theme covers a number of subthemes. Figure 5 provides an overview of these themes and the accompanying subthemes, on which we will elaborate below.

#### 3.2.1. Theme 1: At Home, Alone?

A considerable proportion of the decline in HHC utilization during the first wave of COVID-19, as shown in Figure 1, was caused by the reduced demand for care by non-COVID-19 clients, both new and existing. This was driven by fear of infection, with many *clients opting out of the HHC services themselves* offered by registered nurses, certified nurse assistants, or care assistants. Instead, these clients chose to rely more on self-care and/or on support from informal caregivers, such as family members, friends, or neighbors. For clients who did not opt out of care, in many cases HHC organizations scaled back their service provision, either completely or partially depending on the level of need (this corresponds to what we see in Figure 4). This meant that clients were approached by a district nurse to discuss which services could be provided less frequently, eliminated completely, and/or provided through video calls or informal care. There was thus a *formal needs re-assessment* of all existing clients aimed at scaling back formal care delivery as much as possible. Clients generally felt well informed and actively involved in the decision-making process around scaling back care:


*“I have the impression that scaling back care was discussed properly with the clients and informal caregivers. They really have a say in that, resulting in a genuine dialogue [between the client, informal caregiver, and the nurse about scaling back care] that resulted in a solution.” (client council member D4a)*


Regardless of whether it was driven by personal choice or supply constraints, the scaling back of formal care provision had direct consequences for HHC clients. Clients became more dependent on *self-management and informal care*. Especially for potential new clients, the fear of infection was a reason for relying more on themselves or on informal caregivers, rather than opting to receive HHC. For some existing clients, increased self-reliance and use of informal care has worked out positively and they have continued to receive less HHC or no HHC at all. Encouragement to be more self-reliant was much more readily accepted by clients than before the pandemic, for example, for those who had been receiving the same low-complexity care for years. The aspects above were shared widely across all levels in the focus group interviews. For some existing clients who might have become eligible for admission to a nursing home during the pandemic, the closure of nursing homes to visitors and other non-essential persons (from March 19 2020 onwards) was a reason for them to continue to live at home and receive HHC for longer. According to nurses, most clients experienced no major adverse effects from the changes in HHC provision. However, some informal caregivers reported a higher burden, with the potential risk of becoming overburdened, after formal care had been scaled back.


*“As a district nurse, when you carry out a needs assessment you always look at […] how a client could become more independent, but not every client is open to that. Now [during the pandemic], clients were much more receptive.” (manager A2a)*



*“A few weeks after care was scaled back, we had informal caregivers contacting us saying: ‘I said I could take over [the care for the client], but I am having a tough time’.” (director D1a)*


However, according to all the nurses, for some clients the reduction in formal care (including in other healthcare sectors) led to undertreatment, eventually resulting in an even higher need for HHC. The consequences of limited formal care were felt *particularly by clients with cognitive decline*. Some other significant adverse effects of the pandemic and the associated care restrictions were reported mainly by the nurses and the members of the client councils. As the scaling back of HHC took place at the same time as many other restrictions in the social environment—such as the closure of daycare facilities, restrictions on social contact, and so on—this led to more *loneliness* among clients. 


*“[Due to the COVID-19 pandemic,] the daily routine for clients with dementia has changed, making them more depressed, confused, and lonely. That does not necessarily mean that they need to be admitted to a nursing home, but we are doing a lot to help these clients”. (district nurse B2b)*


#### 3.2.2. Theme 2: Managing Uncertain Demands

Across the HHC organizations studied, for managers, the first wave of the COVID-19 pandemic was an unprecedented crisis that unfolded at great speed. Communication between HHC organizations intensified, and active regional collaboration was initiated. Directors and managers very much valued this sense of solidarity between organizations. Internally, all organizations responded in a *top-down* manner to start with, creating a *crisis team* to bring together high-level management functions, including directors, operational managers, and policy advisors. For managers, communicating with HHC professionals was difficult. This was because they had to work from home and thus felt more distant from HHC professionals. It was also hard to structure the very large amounts of information they received from within and outside of their organization. 


*“HHC professionals couldn’t see the wood for the trees, and neither could we as managers when we were getting 80 e-mails just about ever-changing COVID-19 policies.” (crisis team member C2a)*


Of the many policy decisions made by the crisis team, two were raised the most frequently and discussed in the most detail during the focus group interviews. First, the decision to *scale back regular care to the absolute minimum in anticipation* of massive COVID-19-related demand for care from the hospital sector. Due to the high demand for COVID-19-related hospital care, hospitals were already scaling back planned non-COVID-19-related care, such as minor surgery, leading to a decrease in post-hospital HHC.

In the first wave, the needs of all (regular) HHC clients were re-assessed and subsequently categorized depending on how essential their care needs were (known as the ‘traffic light system’ by some organizations). This meant that essential, medically necessary care—such as palliative care and technical nursing care (e.g., dressing a wound or providing medication)—had to be continued under all circumstances. Other types of care—such as personal care, preventive home visits—were categorized in terms of necessity and alternative forms of care provision were suggested. Examples include teaching an informal caregiver to give a client’s eye drops, decreasing the frequency of showering or washing from three times to once a week, or regular telephone contact to check how a client was doing. The re-assessment of care needs and types of care went hand in hand with a rise in the use of e-health tools and assistive equipment in three out of the four HHC organizations.


*“In the first wave, we had no idea what to expect, so we developed and introduced scenarios for scaling back, because we thought we might not have enough HHC personnel available. But actually the situation wasn’t as bad as expected for us.” (crisis team member C2a)*



*“For clients whose care provision was scaled back, our organization made sure they had weekly telephone contact with them to support them as well as possible.” (client council member B4b)*


The end of the first wave was characterized by a fall in the number of COVID-19 cases. According to the directors, it became clear in hindsight that the scaling back of HHC during the first wave had been too rigorous. HHC organizations learned from the first wave, resulting in much less scaling back of HHC services during the second wave. This was in line with our quantitative findings (i.e., Figure 1, Figure 2, Figure 3 and Figure 4), in which we did not observe any declines as pronounced as during the first wave.


*“Based on the numbers [of infections] in our region, we expected a huge wave of clients [to be discharged from the hospital] who would need HHC. From all sides, we were being told: ‘Get ready for [clients coming out of the] hospitals, scale back your care!’ However, this huge wave never came. […] Looking back, we scaled back more than was strictly necessary, but nobody knew that at the time. […] You assumed that [the need to scale back care] would only last for a few weeks, and that we’d manage it.” (director D1a)*


Second, the role and responsibilities of management in the *limited information and resources available* to protect nurses in the HHC setting was addressed. In the first wave, there was a lack of knowledge and experience within HHC organizations and among clients regarding what COVID-19 was and how to work with COVID-19-infected clients. Clients (suspected of) having COVID-19, therefore, received HHC separately from other regular clients, from special COVID-19 teams. For most organizations, nurses could join the COVID-19 teams voluntarily. During this first wave, HHC organizations had to deal with a severe *shortage of PPE,* such as facemasks and gloves. According to national guidelines, HHC professionals were not supposed to use PPE with all HHC clients, but only when visiting a client infected with COVID-19. Policy advisors and directors of the HHC organizations found themselves in a dilemma: either stick to the national policy—which most of them did—or make their own decisions. For example, some managers indicated that they would sometimes disregard national policy in individual cases, using their own professional insight. Some HHC organizations also tried to purchase extra equipment themselves rather than waiting for action at the national level.


*“[Our HHC professionals] felt extremely unsafe, and there was nothing we could do about it because the resources just weren’t there. […] As a manager, I found that very difficult.” (manager A1a)*



*“At a certain point, we [our organization] bought PPE ourselves because we weren’t getting anything from the regional distribution of equipment. […] That degree of divergence between organizations—that shouldn’t be allowed to happen in my opinion.” (director C1a)*


Since the second wave began, PPE became more widely available again and all professionals were able to wear the equipment when this was considered necessary. This—together with increased knowledge on how to live and work with COVID-19 and the fall in the number of cases—made it possible to return COVID-19-related care to the regular HHC teams. This also fulfilled the need for HHC organizations to implement a longer-term solution than COVID-19 teams.

#### 3.2.3. Theme 3: Being a Nurse in Times of Crisis

On the one hand, the top-down approach in the first wave was generally viewed positively by both managers and nurses because it enabled fast decision making. However, nurses also indicated that it did not allow for their *professional expertise to become an integral part of the decision-making processes* and limited their autonomy in care provision. During the pandemic, nurses functioned as a link between management and clients. They had to communicate decisions on crisis management policies made by management to the individual clients. This was to provide personalized information and avoid unrest among the clients as much as possible. Due to the overall discontent with the top-down approach and the nurses’ central role, most organizations added nurses to the crisis team—and client council members too—to ensure a more bottom-up approach in the second wave.


*“We received feedback from our employees that they felt there was a major gap between them and their managers in the first wave. So in the second wave, […] we introduced all sorts of initiatives for our employees to be more involved in decision making on our policy.” (manager A2a)*


Nurses experienced *heavy workloads* during the pandemic. Particularly those working in the COVID-19 teams, but also nurses in teams where more colleagues were absent due to sickness, for example, worked longer hours than normal (some nurses mentioned 15 h working days) and were unable to take any days off. Nurses in regular HHC teams also faced additional tasks at work due to the scaling back of care in other settings such as hospital care and nursing homes, arranging COVID-19 tests for their HHC clients, or re-organizing schedules due to colleagues who were sick. 


*“I received many phone calls in my own time, from colleagues asking for help or telling me about clients who were infected. […] I was constantly thinking about who was or might be infected, who could come out of quarantine, etc. […] And arranging for clients to be tested for COVID-19 by their GP took up a lot of my time.” (district nurse D3a)*


During the first wave, the *working environment was considered very unsafe* by nurses. According to nurses, their organizations were following national policy guidelines on PPE for much too long, and they should have been doing more to protect their employees. Many nurses and other HHC professionals felt let down by the government and their employers. Nurses were afraid of getting infected and spreading the virus to clients or to their own families. This also led to frequent testing for the virus, which further added to the perceived stress. In addition to the unsafe working environment, the information overload from managers and continuously changing procedures and protocols led to *unrest* among nurses, especially during the first wave. Nurses were not always aware of the latest procedures and protocols, which sometimes affected clients negatively. During the second wave, although more knowledge and PPE were available and changes in procedures and protocols were less frequent, the return of COVID-19-related care to the regular HHC teams increased uncertainty among the nurses once again. Good dissemination of knowledge from the COVID-19 teams to the regular HHC teams and the availability of experts on COVID-19-related care to answer questions were helpful in these cases, and reduced levels of anxiety and uncertainty. Even though the working conditions were considered tough by nurses, they also acknowledged the difficulties that managers and directors must have been experiencing. 


*“As employees, we were obliged to go to work, even if you had a family member at home who had tested positive for COVID-19. I felt pressure because of that.” (district nurse D3a)*



*“[The level of unrest during the second wave] was different in each team. Some teams said: ‘We have PPE now. We know who has tested positive and what to do.’ But other teams still panic a little if a client tests positive, wondering ‘What should we do now?!’” (policy advisor B2a)*


Nurses indicated that their resilience and wellbeing had been negatively impacted by the difficult working conditions and the difficult ethical choices that had to be made when scaling back HHC provision to clients in need. Nurses were and continue to be *emotionally impacted* by these difficulties, especially those who had a high number of sick colleagues in their teams. While in the first wave, nurses were able to soldier on, their resilience by the end of the second wave was felt to be just enough to carry on. The uncertainty around future waves of COVID-19 was putting a strain on all those involved in HHC. Nurses, as well as directors, managers, and clients, wondered if people could cope with the COVID-19 situation any longer. Monitoring long-term effects, which are to be expected, at a personal level would be a valuable exercise.


*“Some of my colleagues believe that they infected clients. They still have that on their mind and it’s a source of stress, and as a result they are currently on sick leave.” (district nurse C3b)*



*“[In the second wave,] the capability to carry the burden [of the COVID-19 pandemic] was lower compared to the first wave, but we all still just got on with it.” (manager A1a)*



*“When I see how everything is going now, I wonder how the next wave will turn out. […] [The HHC organizations] made it through the second wave, but whether it will continue to work and whether they have the resilience to absorb the next hit… I find that a frightening thought to be honest.” (client council member D4a)*


## 4. Discussion

The main findings of our research are that particularly during the first wave of the COVID-19 pandemic (i.e., between March and June 2020), both the demand for and supply of HHC decreased considerably. The different perspectives in the focus group interviews enabled us to understand these changes in light of the responses to and experiences of HHC staff and clients to the COVID-19 pandemic in 2020. Both clients newly eligible for HHC and existing clients opted out of formal care and relied (more) on self-care and informal care. At the same time, HHC organizations reduced the provision of care in anticipation of very high demand for COVID-19-related care from the hospital sector and a lack of PPE. During the period of reduced supply, nurses monitored their clients and adjusted the supply of HHC on the basis of clients’ needs. During the first wave, HHC organizations used a centralized, top-down approach to cope with the unanticipated crisis. After the first wave, HHC organizations granted more professional autonomy to nurses. Additionally, HHC organizations learned lessons from the first wave with regard to scaling back care, making smaller reductions in HHC services during the second wave compared to the first wave. Experiences of working in HHC during the COVID-19 pandemic differed between participants and between waves. While there was a prevailing sentiment of ‘we survived’, there were also fears about how to deal with potential future waves.

We saw the use of HHC decline significantly across the board between March 2020 and January 2021, with the most significant decrease in production occurring during the first COVID-19 wave. While all HHC organizations participating in our study scaled back care in a structured manner, our findings suggest that in retrospect the steps taken were ‘too much, too soon’. Although we do not know what proportion of the reduction in the use of services was due to clients preferring to avoid medical care, this is likely to have been a significant factor. However, based on our focus group interviews, one overarching factor clearly influenced all stakeholders (i.e., managers, nurses, and clients): fear. Similar to our findings, studies in other countries suggest that many clients—and/or their informal caregivers—were unwilling to allow HHC professionals into their homes during the first wave of the pandemic because they were so concerned about being infected [16]. At the same time, the fear of unprecedented demand for HHC from COVID-19 patients discharged from hospital led managers in Dutch HHC to drastically scale back service provision. However, this fear ultimately proved unfounded, and many focus group participants criticized the timing and extent of the scaling back of HHC services, in retrospect. Future crisis management protocols should enable managers to scale back where needed, responding rapidly and flexibly, rather than in anticipation of uncertain demand.

There was an expectation that service levels might remain at a lower level after the reduction in HHC because of clients’ increased self-reliance; however, our findings suggest that has not happened. As in other studies [2], service provision was almost back to its original levels by the time of the second wave of the pandemic. It seems that informal caregivers took on most care activities during the first wave. Research among informal caregivers of patients with dementia living at home in Norway, for example, also showed that almost 70% of relatives reported an increase in their care responsibilities [4]. The major staff shortages within HHC [17]—which were exacerbated further by the COVID-19 pandemic due to people’s fear of infection [3] and high rates of sick leave [18]—underline the importance of shifting services from HHC professionals to informal caregivers or encouraging clients to become more independent. However, the increase in the intensity of informal care provision also resulted in a higher burden for informal caregivers. Reviews of the impact of the COVID-19 pandemic on informal caregivers have indicated that it had a major impact on informal caregivers—including higher stress levels, pain, depressive symptoms, sleep problems, and greater social isolation—and 72% of the informal caregivers in the US are feeling more burned out than ever [19,20]. Additional research is therefore needed to learn more about which formal care services provided by HHC professionals can be sustainably substituted by self-care or informal care, and for which clients, under which circumstances.

There are widespread reports of increased stress, burnout, and mental health problems among healthcare professionals caring for COVID-19 patients, including depression, insomnia, and post-traumatic stress disorder [21,22,23]. Studies based on survey data have attributed these problems to the fear of infection, fear of transmitting the virus to others, excessive workloads, and social stigmatization [24]. Indeed, the district nurses in our focus group reported significant anxiety around infection, exacerbated by the lack of proper PPE. As frontline workers whose profession involves significant physical contact, nurses felt frustrated by the lack of protection of their own health and safety provided by their employers, and—consequently—the lack of protection of their clients’ health and safety too. They reported heavy workloads, high absenteeism due to illness, and long working hours. Despite these dire circumstances, district nurses went to great lengths to offer meaningful support to their clients, either in the homes or from a distance, to prevent potentially irreversible declines in their health. It is essential that we develop and implement interventions to help nurses cope with the resulting mental and physical problems, both in order to control the pandemic and to ensure the long-term sustainability of high-quality HHC. Future research, for example a systematic literature review on certain supporting interventions and/or qualitative research with HHC professionals, might be needed to gain more insights into how the HHC professionals can be supported in their work and experiences during and after a crisis. Moreover, HHC professionals should play a central role in crisis management to ensure that their sustainability and well-being are taken into account in decision making. 

Additionally, decisions regarding the discontinuation of care cannot be taken on the basis of a one-size-fits-all approach and the expertise of district nurses should be utilized if we are to prevent adverse outcomes for vulnerable and already disadvantaged populations of HHC clients [25]. It seems promising, in this sense, that nurses were given a formal role in the crisis teams set up by all the participating HHC organizations following the first wave of the pandemic. In that respect, the HHC organizations in our study were already using the professional expertise of district nurses in scaling back services: nurses re-assessed their clients’ needs and made informed decisions on whether discontinuation was possible. Our quantitative findings suggest that—at least from a medical perspective—this was a success: service use fell the most in subgroups of clients with relatively low care needs. Still, it is important to examine whether existing health inequalities in the HHC population were exacerbated due to the discontinuation of service resulting from the decisions made by clients or staff. Recent research from the UK suggests that the government’s approach to the lockdown predominantly impacted women, ethnic minorities, and those with chronic illnesses, thus reducing access to care for groups that were already vulnerable [26].

Finally, although telehealth has been described as a potentially valuable tool for enabling HHC visits without exposing clients and nurses to the risk of COVID-19 infection, our participating organizations turned to a simpler solution in order to maintain contact with vulnerable clients: phone calls. A recent study in the US found that a phone-based outreach program during the COVID-19 pandemic is a relatively simple intervention with a potentially far-reaching consequences [27]. It facilitated clinical assessment and intervention in at-risk groups and enabled the delivery of patient education and—perhaps most importantly—social connection for vulnerable and emotionally isolated clients and families. For older adults living at home who are unfamiliar with e-health and digital forms of communication, phone calls are often more accessible. An initial, small-scale evaluation of a similar program in which volunteers called rather than healthcare professionals showed promising effects on participants’ social and emotional well-being, as well as overall self-reported health [28]. Continuing such telephone-based interventions, even after the current pandemic, could be an effective way of supporting the health and well-being of older adults living at home. 

The strength of our study was its combining of quantitative data with in-depth focus group discussions that included a range of perspectives—including nurses, management, and clients—by studying a large amount of quantitative client data from four organizations. First, we were able to obtain feedback from those organizations and direct the discussion during the focus group interviews. Although we believe that the HHC organizations involved in this research are representative of the Netherlands, and only minor differences were noted between organizations in the focus group interviews, we cannot exclude the possibility that other organizations may have responded differently to the crisis. We must, therefore, be cautious regarding the generalizability of these results. A further limitation of our study is the relatively short timeframe in which it was conducted, especially given that the pandemic is still ongoing. It would be interesting to investigate the longer-term effects of the COVID-19 pandemic on HHC. Future research would therefore be valuable, in which possibly other quantitative analyses are applied on a more extensive dataset of, for example, multiple years, to assess these effects. However, we believe that the insights that we have gained into the first and second waves provide valuable lessons for future crises and for subsequent waves in this pandemic.

## 5. Conclusions

This study shows that HHC organizations had to deliver HHC to vulnerable clients in difficult times during the first two waves (March 2020–January 2021) of the COVID-19 pandemic. In retrospect, HHC organizations scaled back the delivery of HHC too drastically in the first wave. In the future, more responsive crisis management protocols should be made available that scale back care only if this is truly needed. Nurses monitored and adjusted the provision of HHC on the basis of individual needs. However, future research will need to demonstrate whether certain HHC clients were particularly impacted by the scaling back of HHC and for which groups self-care, informal care, or e-health solutions might be appropriate. Notwithstanding all the efforts to provide the best HHC possible tailored to clients’ needs, if those providing the care do not receive sufficient support, there is nobody else to act during times of crisis. Developing and implementing interventions to help HHC professionals cope with the effects of a crisis is therefore essential for the long-term sustainability of high-quality HHC.

## Figures and Tables

**Figure 1 ijerph-19-02252-f001:**
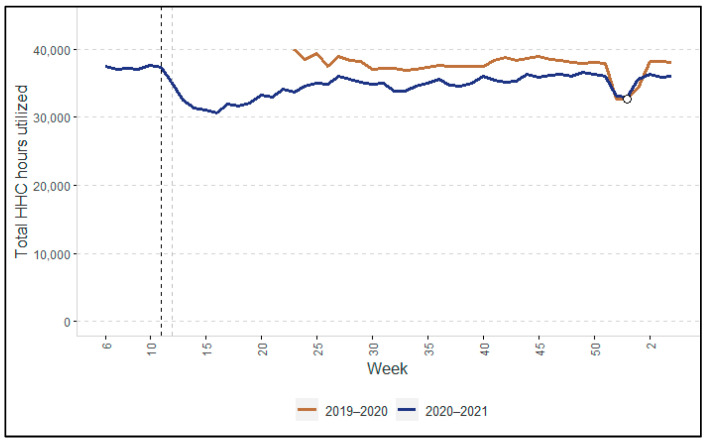
Total hours of HHC services provided in 2020–2021, compared with the previous year (2019–2020). Note: Because the year 2020 had 53 weeks rather than 52 weeks, we added a datapoint (the white dot) for week 53 for 2019 using the value of week 52.

**Figure 2 ijerph-19-02252-f002:**
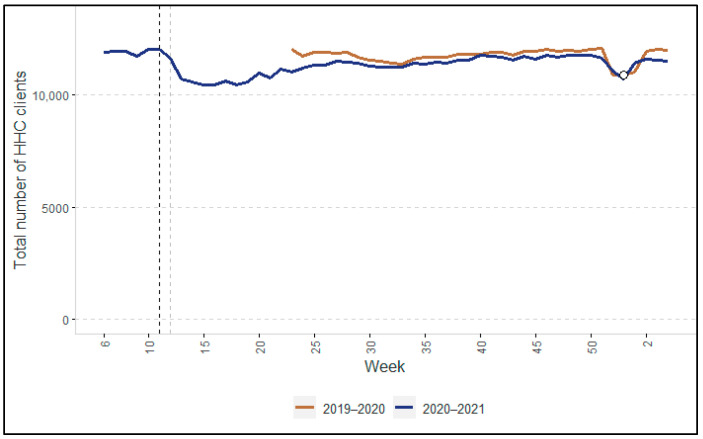
Total number of HHC clients in 2020–2021, compared with the previous year (2019–2020). Note: Because the year 2020 had 53 weeks rather than 52 weeks, we added a datapoint (the white dot) for week 53 for 2019 using the value of week 52.

**Figure 3 ijerph-19-02252-f003:**
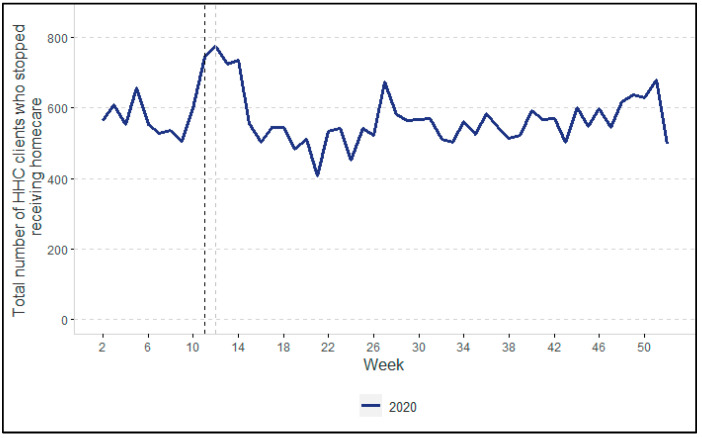
Number of HHC clients who stopped receiving homecare in 2020.

**Figure 4 ijerph-19-02252-f004:**
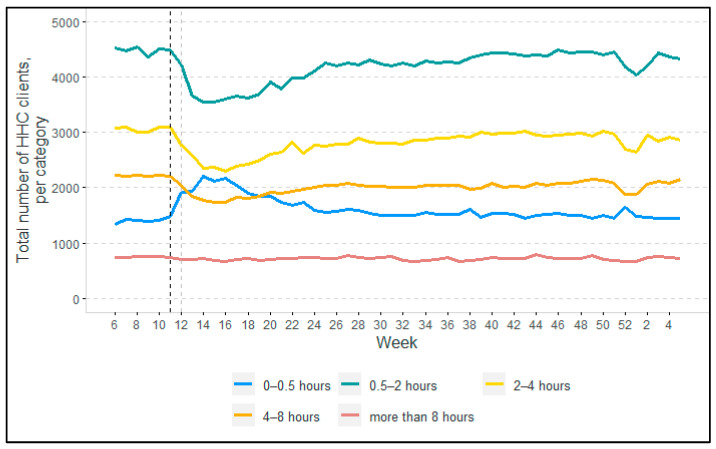
Intensity of HHC in 2020–2021, according to the number of HHC clients per category of hours of HHC utilized.

**Figure 5 ijerph-19-02252-f005:**
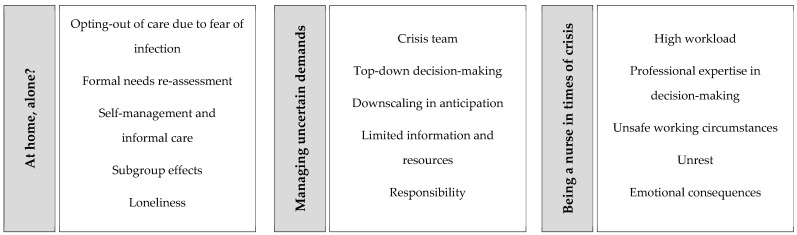
Themes (grey boxes) and sub-themes (white boxes) derived from the qualitative analyses.

**Table 1 ijerph-19-02252-t001:** The number and positions of focus group interview participants, per focus group.

Stakeholder Level			
Strategic	Tactical	Operational	Client
N = 4	N = 6	N = 11	N = 7
3 directors 1 manager	2 managers 2 policy advisors 1 crisis team member 1 team leader	11 district nurses	7 client council members

## Data Availability

The data used in this study can be made available by the corresponding author on reasonable request.

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
