# Peer review of "In the Eye of the Storm: A Quantitative and Qualitative Account of the Impact of the COVID-19 Pandemic on Dutch Home Healthcare"

_ijerph, 2022, doi:10.3390/ijerph19042252_

Round 1

Reviewer 1 Report

Thank you for the opportunity to review this manuscript, which examines the impact of the COVID-19 pandemic on home healthcare with mixed methods. I think this study has significance, but here are suggestions to improve the manuscript

1. The title should include the Netherlands to indicate the area of study.
2. There should be more desciptions about the data (e.g., the number, country, whether the survey was online) in the abstract.
3. Although I appreciate that the authors used mixed methods, I think the quantitative should be more elaborated. The current quantitative analysis is too descriptive. If the authors have other possible variables (e.g., age, gender, ethnicity) to predict home healthcare, I suggest they run OLS regression or other multivariate analysis. 
4. The Netherlands context should be explained in the early introduction. For example, what are the needs of home healthcare? Is the private sector dominant to provide home healthcare? Does the government provide any subsidies?
5. Methods: This section needs more descriptions. For example, do all clients mean patients or nurses? How many participants were included? What were the inclusion and exclusion criteria?
6. Table 1 is too complicated.
7. Figures: The authors could adjust the scale of the Y-axis.
8. I think it would be nice if the authors include suggestions for future studies. 

Author Response

Please see the attachment for our point-by-point response to the reviewers' comments.

Reviewer 2 Report

Although the topic seems to be interesting, but the method and the analysis  of data suffer from important limitations.

In quantitative part, Authors just reported the crude rates and no adjustment and/or specific analysis to show the significance of the changes during the time, has been performed.

Usually in this case, Poisson regression analysis between number of demands or number of hours of care received, as dependent variable, should be performed by the independent time and number of covid-19 cases in the population with some specific modeling. This can reflect the correct correlations and associations.

Some other approaches like changes according to the baseline and/or changes to the average can also be used to show the significance of change during the time. Although authors mentioned some calculations related to the last approach but did not use that as a main approach in analysis.

Another important change point in diagrams is related to the end of the year 2020 and start of the year 2021, which needs more investigation and comparison in significance to the decrease rate in the first wave of the disease observed in the community.

In qualitative focus group part, the objects are not clear, some findings have been presented and discussed but at least two points have been remained unclear, the way authors reached to data saturation and the practical implications of each finding, specifically for next research.

In present form, data can be published only in a short communication or letter to editor.

Decision: Reject.

Author Response

(The authors gave the same response as above.)

Round 2

Reviewer 1 Report

The authors revised the manuscript well based on the reviewer's comments. 

Author Response

Response to the comments of reviewer 1:

  • The authors revised the manuscript well based on the reviewer's comments.

Our response: We would like to thank reviewer 1 for the compliment and the effort put into the reviewing process of our manuscript.

Reviewer 2 Report

English language editing is not performed thoroughly across the manuscript.

The main question in the quantitative part of the study is the effect of the covid-19 intensity on changes in the HHC provision. If there is no data about number of covid-19 cases or mortality per day in the model conjunct with the time, how the changes can be considered in association with covid-19? I brought the Christmas example for showing the fact that for proving the association we shall have some other data regarding to the covid-19 rather than the pure time. 

In case of the qualitative part of the study, it can explain the path, covid-19 might have caused the changes but first the quantitative one should be trustable.

Discussion part and conclusion needs changes if authors intend to present the manuscript in this form rather than a short communication.

Multivariate analysis should be a part of the study not just mentioned in the supplements and should be a basis for discussion.

Decision: Major revision.

Author Response

Response to the comments of reviewer 2:

  • Moderate English changes required. English language editing is not performed thoroughly across the manuscript.

Our response: From this comment, we do not know where in the manuscript the reviewer would like to see changes. Also, we did have our manuscript reviewed in detail by a professional English language proof reader. Therefore, we would like to leave the decision on the need for further improvement up to the editor.

  • The main question in the quantitative part of the study is the effect of the covid-19 intensity on changes in the HHC provision. If there is no data about number of covid-19 cases or mortality per day in the model conjunct with the time, how the changes can be considered in association with covid-19? I brought the Christmas example for showing the fact that for proving the association we shall have some other data regarding to the covid-19 rather than the pure time.

Our response: It seems that some misunderstanding occurred regarding our first study aim. Based on the reviewer’s comment, if the aim would be to study changes in HHC provision in relation to the intensity of COVID (i.e. the number of COVID-cases), data on for example the number of COVID-cases would seem to be lacking. However, the way the reviewer describes our main question is not in line with how we formulated the aims in our manuscript. In the Introduction, we stated: “Our first aim was to describe how Dutch HHC organizations responded to the COVID-19 pandemic (March 2020 – January 2021), in terms of changes in the volume and intensity of HHC services, and the crisis management policies implemented.” Thus, we were not particularly interested in the effect of the intensity of COVID (i.e. the number of COVID-cases) on HHC use, but rather of the COVID-pandemic in general. We do agree that it is important to carefully interpret the observed trends. We chose to exploit the value of mixed-methods research for this purpose, instead of using additional data. That is because mixed-methods research permits a more complete utilization of data than do quantitative data alone. However, we do recommend performing additional research using other data. This recommendation was added based on the reviewers’ comments previously.

  • In case of the qualitative part of the study, it can explain the path, covid-19 might have caused the changes but first the quantitative one should be trustable.

Our response: We elaborated on the suggestions made for the quantitative part of the study in our response to the previous comment. Regarding the qualitative part of our study: in the focus group interviews, we indeed discussed potential explanations for the changes in HHC use with the participants. This was in line with our second study aim: “Second, we aimed to understand these responses [to the COVID-19 pandemic] at all levels of HHC – i.e. strategic, tactical, operational and client levels – and to learn more about their experiences.”

  • Discussion part and conclusion needs changes if authors intend to present the manuscript in this form rather than a short communication.

Our response: Our viewpoint on this matter has not changed in comparison to the response to the reviewer’s comment previously. According to us, the combination of descriptive data and the qualitative data that support the understanding of the effects of the COVID-pandemic on HHC, requires an extensive description of e.g. the methodological steps taken and the theoretical discussion that follows from our results. A short communication or letter to the editor would not lend itself for this purpose. We are sorry that the reviewer did not change his/her mind regarding the value of our study as a research article despite our revisions made and explanations given.

  • Multivariate analysis should be a part of the study not just mentioned in the supplements and should be a basis for discussion.

Our response: A key aspect of a mixed-methods study is that it is not simply the gathering of quantitative and qualitative data. Instead, it is an integration of quantitative and qualitative data. With the explanatory mixed-methods design of our study, we intended the quantitative outcomes to be supportive for the discussion in the focus groups. The data used for the additional regression analysis were, however, not yet available during the focus groups, as the analysis was performed afterwards based on the reviewers’ comments. Therefore, considering the additional analysis and findings as part of our explanatory mixed-methods design would not be appropriate due to impossibility of discussing these during the focus groups. Therefore, our viewpoint remains that additional analyses which were not discussed during the focus group interviews should not be considered part of the explanatory mixed-methods design. We believe that including the additional analysis as supplementary materials is therefore the most appropriate solution.